

# Public perspectives and media reporting of wolf reintroduction in Colorado

Rebecca Niemiec[1], Richard E.W. Berl[1], Mireille Gonzalez[1], Tara Teel[1], Cassiopeia Camara[1], Matthew Collins[1], Jonathan Salerno[1], Kevin Crooks[2], Courtney Schultz[3], Stewart Breck[2,4] and Dana Hoag[5]

[1] Department of Human Dimensions of Natural Resources, Colorado State University, Fort Collins, CO, USA
[2] Department of Fish, Wildlife, and Conservation Biology, Colorado State University, Fort Collins, CO, USA
[3] Department of Forest and Rangeland Stewardship, Colorado State University, Fort Collins, CO, USA
[4] USDA National Wildlife Research Center, Fort Collins, CO, USA
[5] Department of Agricultural and Resource Economics, Colorado State University, Fort Collins, CO, USA

Corresponding author
Rebecca Niemiec,
rebecca.niemiec@colostate.edu

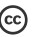

## ABSTRACT

In the state of Colorado, a citizen ballot initiative to reintroduce gray wolves (*Canis lupus*) is eliciting polarization and conflict among multiple stakeholder and interest groups. Given this complex social landscape, we examined the social context surrounding wolf reintroduction in Colorado as of 2019. We used an online survey of 734 Coloradans representative in terms of age and gender, and we sampled from different regions across the state, to examine public beliefs and attitudes related to wolf reintroduction and various wolf management options. We also conducted a content analysis of media coverage on potential wolf reintroduction in 10 major daily Colorado newspapers from January 2019, when the signature-gathering effort for the wolf reintroduction initiative began, through the end of January 2020, when the initiative was officially added to the ballot. Our findings suggest a high degree of social tolerance or desire for wolf reintroduction in Colorado across geographies, stakeholder groups, and demographics. However, we also find that a portion of the public believes that wolves would negatively impact their livelihoods, primarily because of concerns over the safety of people and pets, loss of hunting opportunities, and potential wolf predation on livestock. These concerns—particularly those related to livestock losses—are strongly reflected in the media. We find that media coverage has focused only on a few of the many perceived positive and negative impacts of wolf reintroduction identified among the public. Our findings highlight the need to account for this diversity of perspectives in future decisions and to conduct public outreach regarding likely impacts of wolf reintroduction.

## INTRODUCTION

Predator management and reintroduction continues to be one of the most contentious conservation issues in the American West, and wolves in particular elicit significant social

conflict (*Bruskotter, 2013*; *Wilson, 1997*). To some, the reintroduction of wolves symbolizes a chance to make amends with wilderness, while, for others, wolves are seen as a threat to livestock, hunted ungulate populations, and rural livelihoods (*Kellert et al., 1996*; *Slagle et al., 2019*; *Wilson, 1997*). The debate over wolves is often about much more than just wolves, as wolves serve as a surrogate or representative issue that reflects broader societal-level conflicts, such as urban versus rural values or state versus federal roles in land and wildlife management (*Nie, 2001*; *Wilson, 1997*). Understanding public attitudes, beliefs, and discourse related to wolves is therefore important not only to guide management decisions about wolves, but also to inform broader discourse on wildlife conservation and humans' relationship with their environment (*Wilson, 1997*). Scholars have argued that wolves can function as a "particularly powerful barometer of changing and conflicting attitudes toward wildlife" (*Kellert et al., 1996*).

Public attitudes in the western United States are generally positive towards wolves (*Bright & Manfredo, 1996*; *Bruskotter, Schmidt & Teel, 2007*; *George et al., 2016*; *Williams, Ericsson & Heberlein, 2002*), although these attitudes can vary significantly by experience with or proximity to wolves, by interest groups, and by demographics. *Houston, Bruskotter & Fan (2010)*, for example, found that the anticipated presence of wolf populations in a state can increase negative discourse about wolves in the media, which may promote changes in attitudes among people with little knowledge of the topic. The authors also found that states with new populations of wolves had significantly more negative discourse than states with permanent wolf populations. Relatedly, other research shows that people living inside wolf territories have more negative attitudes towards wolf conservation than people living just outside (*Karlsson & Sjöström, 2007*). Attitudes toward wolves have also been found to negatively correlate with a person's age, participation in hunting, direct experience with wolf predation, and ranching and farming occupations, and positively correlate with education and income (*Ericsson & Heberlein, 2003*; *Williams, Ericsson & Heberlein, 2002*; *Sponarski et al., 2013*).

The majority of existing studies on public perceptions toward wolf reintroduction and management in the western United States have focused on contexts where reintroduction was led by the federal government under the Endangered Species Act of 1973 or where wolves returned through natural migration (*Williams, Ericsson & Heberlein, 2002*). In the state of Colorado, by contrast, there is the possibility of wolf reintroduction being mandated via a November 2020 citizen ballot initiative. Initiative 107 is being promoted by numerous conservation organizations and would mandate that Colorado Parks and Wildlife, the state wildlife agency, reintroduce wolves into the state. On January 6th, 2020, the Secretary of State's Office qualified Initiative 107 for the 2020 ballot. Previous surveys suggest that such a measure would pass, assuming public opinion has remained consistent over the past two decades. A 1994 mail survey found that 70.8% of Colorado residents were supportive of wolf reintroduction (*Pate et al., 1996*), while a 2001 phone survey found that 66.0% of Colorado residents were supportive (*Meadow et al., 2005*).

It is possible, however, that attitudes, beliefs and discourse among the public and media about wolf reintroduction have changed and also may be strongly influenced by the unique

political context of a ballot initiative. For example, the ballot initiative raises the question of whether values of the general public, expressed directly through a popular vote, should have more influence in decision-making about wildlife, and whether these values should predominate over those of traditional interest groups (e.g., hunters and anglers) typically included in decision-making by state wildlife agencies (*Manfredo et al., 2017*). Ballot measures are increasingly being used by stakeholder groups with diverse values seeking to influence wildlife management. For example, *Minnis (1998)* reported on six different citizen-led ballot initiatives in 1996 seeking to reduce or ban hunting and trapping. Coloradans voted in favor of a controversial ballot measure banning wildlife trapping in the same year, subverting recommended plans by the state wildlife agency (*Manfredo et al., 1999*). Studies suggest that the rise of citizen-led ballot initiatives is a product of a culture clash, driven by a growing percentage of the U.S. population with more mutualistic wildlife values (i.e., emphasizing beliefs that wildlife should have rights like humans and human activity should be limited for the benefit of wildlife) that contrast with the traditional utilitarian interests that have predominated in state agency decision-making (i.e., hunting and fishing; *Manfredo et al., 2017*). These traditional interest groups have also increasingly utilized ballot initiatives to try to protect hunting rights, in an effort to "fight back" amidst social change (*Manfredo et al., 2017*).

The possibility of wolf reintroduction in Colorado thus provides a case study to inform the broader question of how wildlife can and should be managed in light of shifting societal values. Given the political context of the ballot initiative and the importance of the wolf issue in understanding public attitudes toward wildlife management more broadly, we examined the social context surrounding wolf reintroduction in Colorado in advance of the 2020 ballot initiative. In particular, we examined public beliefs and attitudes related to wolf reintroduction and various wolf management options should wolves be reintroduced. Furthermore, given literature suggesting that media portrayals of wolves may not align with public attitudes but may still influence those attitudes (*Houston, Bruskotter & Fan, 2010*), we sought to compare the discourse around wolf reintroduction in the media with public beliefs and attitudes.

We focused on four research objectives aimed at determining: (1) the level of public support for wolf reintroduction in Colorado, including analyses by demographics, social identities, and geographic regions of the state; (2) the level of support for various wolf management options (e.g., lethal and nonlethal control, compensation for livestock loss), should wolves be reintroduced; (3) Coloradans' perceptions of the impacts of wolves on their livelihoods should wolves be reintroduced; and (4) how wolf reintroduction is being framed by the media and whether this framing is in line with public perspectives. To address these objectives, we conducted a statewide survey of Coloradans and a review of media articles on the potential wolf reintroduction between January 2019 and January 2020.

## MATERIALS AND METHODS

To examine our first three objectives, we conducted a statewide survey which built upon measurement methods developed for prior research on Coloradans' attitudes towards

wolves and wolf reintroduction (*Bright & Manfredo, 1996*; *Pate et al., 1996*). To examine our fourth objective, we collected articles from the digital archives of the top 10 Colorado daily newspapers by circulation (*Agility PR Solutions, 2019*).

## Statewide survey

We adopted a stratified sampling approach to obtain a representative sample of Colorado residents, using three geographic regions, age, and gender categories as strata. All participants were at least 18 years of age and were recruited in August 2019 using Qualtrics (Provo, UT), a commercial sampling firm with a licensed online survey platform.

We set a minimum target sample size of 200 participants for each region in Colorado, including the Front Range (11 counties), Western Slope (35 counties), and Eastern Plains (18 counties), allowing for sufficient samples (estimates within ±7% at the 95% confidence level) for comparisons across the different regions. Our region classifications were adopted from *Teel, Bright & Manfredo (2003)*. We were particularly interested in ensuring adequate sample size for the less populous Western Slope, given this is the region where wolves would most likely be restored and find suitable habitat, and thus impact residents more directly (*Carroll et al., 2003*). We also stratified our sample by age and gender to mirror the *American Community Survey (ACS) (2017)* 5-year (2013–2017) estimates retrieved from the Social Explorer database (Table S1). To reflect the 2017 ACS, our target stratification consisted of equal gender representation and equal distribution of the sample across the following age categories: 18–34, 35–54, and 55 years and older (Table S1).

Upon providing Qualtrics the target sample sizes, the firm then recruited and screened potential respondents from existing pools of online survey-takers contacted through panel partners. Qualtrics asked each potential respondent if they would be willing to participate in a survey; to avoid selection bias, they were not told about the topic of the survey. Potential respondents were incentivized to participate in different ways, depending on which panel they were recruited from; for example, panel partners compensate using monetary incentives, game points, gift cards, or other prizes. We first conducted a soft launch of the survey, in which we obtained 50 pilot responses. We checked responses to open-ended questions to ensure participants' answers reflected that they had read the prompt, and checked responses to closed-ended questions for satisficing (i.e., participants selecting the same responses without considering the questions; *Krosnick, Narayan & Smith, 1996*). Final data collection then proceeded until target sample sizes across strata were met. Qualtrics removed respondents who did not pass a speed check (i.e., took the survey too quickly); who did not answer "yes" to screener questions (asking if they would agree to provide thoughtful and honest answers); or who did not provide informed consent. This led to a total of 734 valid responses.

We obtained 365 responses from the Front Range (49.73% of sample), 277 from the Western Slope (37.74%), and 92 from the Eastern Plains (12.53%) (Fig. 1). We received more than the 200 desired target responses from the Front Range and Western Slope because, once these quotas were reached, the Eastern Plains quota was still not reached and recruitment therefore remained open to try to increase sample size in that region. To accurately represent survey responses across the entire state, it was necessary to weight

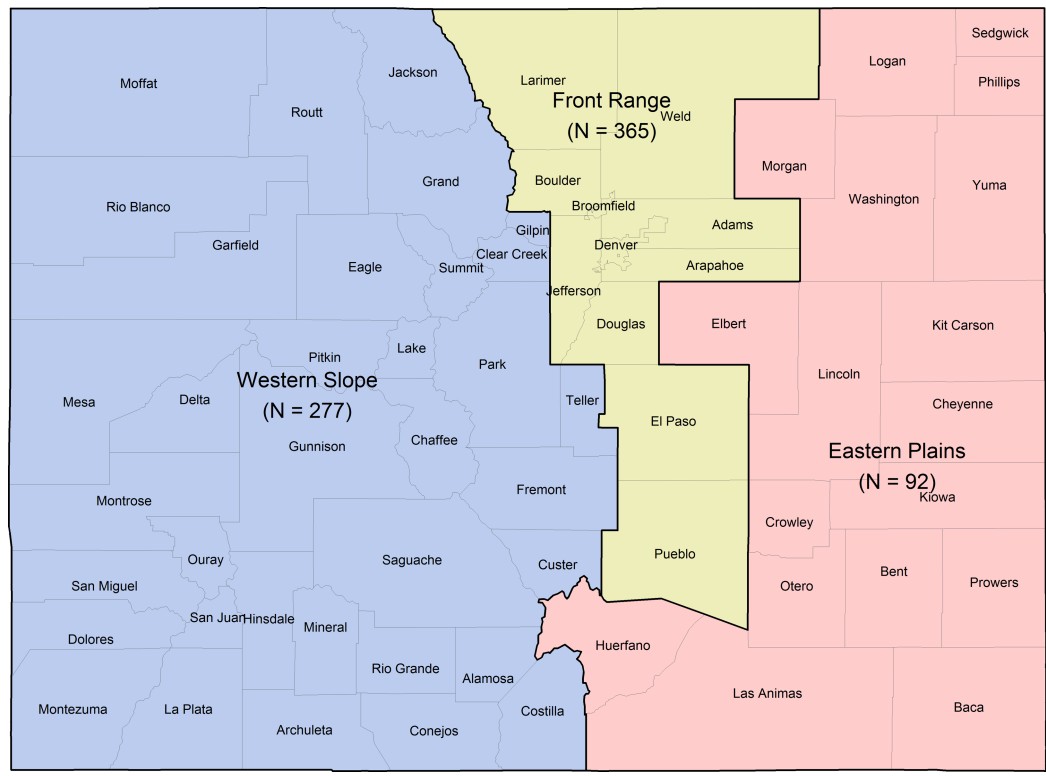

**Figure 1 Colorado counties included within each stratified sampling region and final sample sizes: Western Slope ($n$ = 277), Front Range ($n$ = 365), and Eastern Plains ($n$ = 92).**

the data to reflect the actual population distribution of the three regions for state-level reporting (*Vaske, 2008*). Through our weighting procedure, we made our sample representative of the following geographic population distribution obtained from the ACS (2017): 82.56% of the population in the Front Range, 14.12% of the population in the Western Slope, and 3.31% of the population in the Eastern Plains. The weighting factor for each region was calculated as the population percentage in each region from the 2017 ACS divided by the sample percentage in each region (see Tables S1 and S2 and Supplemental Materials for more information and numbers used for weighting). A full table of weighted and unweighted proportions and weighted confidence intervals is given in the Supplemental Materials (Table S3); weighted and unweighted values were similar.

A limitation that should be considered when interpreting our results is that, because we used an online survey platform, it is possible that our sample may be biased towards individuals with high technology awareness (*Ranchhod & Zhou, 2001*) who are more comfortable with online surveys. *Keeter & McGeeney (2015)* suggest, however, this bias may be small. Past statewide surveys of Coloradans' perspectives towards wolf reintroduction have been conducted by mail (*Pate et al., 1996*) and phone (*Meadow et al., 2005*); thus, it is possible that differences in findings across surveys may reflect differences in survey recruitment rather than respondent attitudes over time.

Despite this limitation, however, we believe our sampling strategy retains a high degree of validity because our survey respondents were recruited independently of their interest in and knowledge of the survey topic, and our recruitment and data weighting procedures were designed to obtain a representative sample of the Colorado population with regard to age, gender, and geography (*Wang et al., 2015*). Furthermore, a growing number of studies are using online platforms for data collection due to increasing technology awareness among the U.S. population (*Ryan & Lewis, 2017*) and given low response rates that are increasingly a challenge for mail and phone surveys (*Keeter et al., 2017*; *Stedman et al., 2019*). Finally, research on alternative survey methodologies indicates that some of the same limitations (e.g., response bias, violations of probability theory) associated with online panel surveys apply to other commonly used survey sampling techniques (*Rivers, 2013*).

## Measurement

The survey included a variety of questions on voting intentions, attitudes and beliefs, experiences, and behaviors related to wolves, wolf reintroduction, and wolf management in Colorado (adapted from *Pate et al. (1996)* and other studies, as described below; see Supplemental Materials for all survey questions reported on). Additionally, we measured identification with various interest groups (e.g., hunters, ranchers, gun rights advocates, wildlife advocates, conservationists), building on *Bruskotter, Vaske & Schmidt (2009)* and *Slagle et al. (2019)*, who found that identification with interest groups influences perspectives towards wolf conservation and management. Identification with interest groups was measured as a four-point scale, ranging from identifying "not at all" to identifying "a great deal." We also measured demographics (e.g., income, age, gender, pet ownership) and location of residence. Final survey and administration procedures were approved by Colorado State University's Institutional Review Board (protocol #19-8942H). Informed consent was obtained from all survey participants; in particular, participants were given the consent language at the beginning of the survey and were then asked if they agreed to participate. If they agreed, they were moved on to the rest of the survey questions; if they did not, their survey was ended. Quantitative analyses were performed and figures were produced using the R statistical environment (version 3.6.1). The data, codebook, and commented R code are provided in full as Supplemental Files for this article.

To address our first two research objectives, we report descriptive statistics on voting intentions, acceptability of various management options, and how intentions varied by demographics, interest groups, and geography. Voting intention was measured by asking "If you were given the opportunity to vote for or against reintroducing the gray wolf into Colorado, how would you vote?", with responses of either for or against (*Pate et al., 1996*). To identify public acceptance of management practices, subjects were asked "If wolf reintroduction were to occur and wolves became reestablished in Colorado, is it acceptable or unacceptable in the future for wildlife management agencies to…"; followed by six potential strategies for participants to rate on a 7-point scale from highly

unacceptable to highly acceptable. These strategies were adapted from *Dietsch et al. (2011)* and included: (1) limit the number of wolves if they cause declines in deer and elk populations in certain areas; (2) capture and lethally remove a wolf if it is known to have caused loss of livestock; (3) compensate landowners for loss of livestock caused by a wolf; (4) use a portion of state hunting and fishing license dollars to compensate landowners for loss of livestock caused by a wolf; (5) use a portion of state tax dollars to compensate landowners for loss of livestock caused by a wolf; and (6) allow a recreational hunt of wolves once they have reached a certain population size that exceeds recovery goals.

To address the third research objective, we report results from the following two survey questions: (1) "To what extent do you feel that wolves would have a direct impact on your livelihood or quality of life?", measured on a 7-point scale, ranging from strong negative to strong positive impact; and (2) "Briefly describe why you feel wolves would negatively/positively impact your life." The second question was only asked of those who indicated that wolves would impact them in their response to the first question. The second question was open-ended, so we used a thematic content analysis (*Braun & Clarke, 2006*) to categorize the most common responses shared among individuals who believed wolves would negatively or positively impact their life. Thematic codes were developed through an iterative process. A three-person team reviewed the survey responses independently and identified a total of 18 themes. Two raters then individually coded all survey responses to these themes, resulting in 83.6% rating concurrence. All non-concurrent responses were examined until 100% concurrence was attained.

## Media analysis

To examine our fourth objective, we collected articles from the digital archives of the top 10 Colorado daily newspapers by circulation (*Agility PR Solutions, 2019*): The Denver Post, Colorado Springs Gazette, The Pueblo Chieftain, Grand Junction Daily Sentinel, Boulder Daily Camera, Fort Collins Coloradoan, Daily Times-Call, Greeley Tribune, Lakewood Sentinel, and Loveland Reporter-Herald. We used the keyword "wolf" to search for articles that were published from January 2019 (when the ballot initiative for wolf reintroduction was announced) through January 2020 (when the announcement was made that enough signatures had been gathered to put the initiative on the ballot). We removed any articles that did not explicitly discuss wolf reintroduction or that were opinion pieces or letters to the editor. This search resulted in 35 articles. We used the thematic codes from the analysis of the open-ended survey responses on perceived positive and negative impacts of wolf reintroduction to code the media articles, because we were specifically interested in whether the media coverage of wolf reintroduction portrayed the diversity of perceived potential impacts we identified through the survey. Two of the three coders who had participated in the coding of open-ended survey responses coded each media article (codes are reported in Table S4 of the Supplemental Materials). This coding process allowed us to examine whether the media reported the same potential impacts from wolf reintroduction reported by the public.

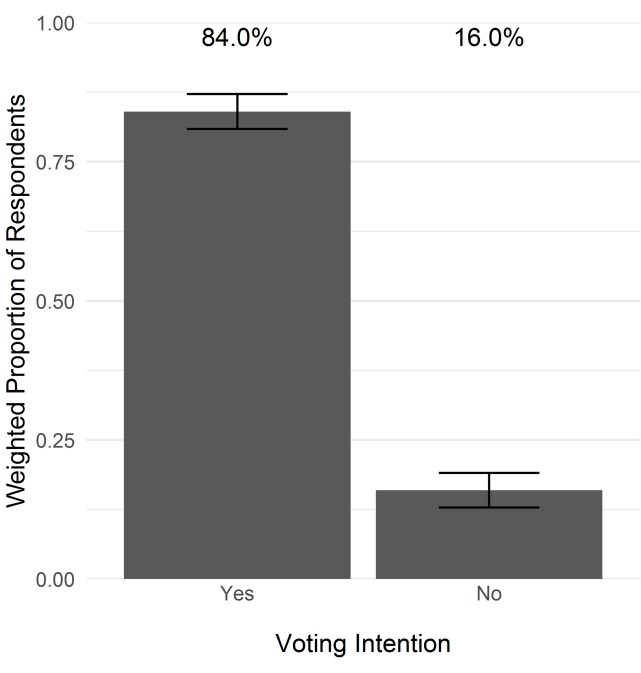

**Figure 2 State-wide voting intentions related to wolf reintroduction in Colorado.** Bars depict the proportion of responses in favor of reintroduction, weighted to represent state demographics, with 95% confidence intervals.    

## RESULTS

### Level of public support for wolf reintroduction

Overall, 84.0% of Coloradans reported intention to vote for wolf reintroduction, while 16.0% reported intention to vote against (Fig. 2). Voting intentions were similar across the different regions of Colorado: 84.9% of Front Range residents, 79.8% of Western Slope residents, and 79.3% of Eastern Plains residents would vote for wolf reintroduction (Fig. 3A). The proportion that would vote for wolf reintroduction was relatively similar among residents in cities, towns, or rural areas and individuals with and without children (Figs. 3B and 3C). Pet owners were more likely to vote for wolf reintroduction (88.3%) than non-pet owners (76.4%; Fig. 3D). Voting intentions were broadly consistent across demographic categories, including gender, age group, income, and education (Figs. 3E–3H). Voting intentions were consistently supportive of wolf reintroduction (>80%) among those who identified (i.e., slightly, moderately, or strongly identified) and those who did not identify as gun rights advocates, property rights advocates, hunters, and ranchers (Fig. 4). Support remained above 80% among those who slightly and moderately identified as ranchers and hunters, but was lower among those who strongly identified as ranchers and hunters (69.5% and 66.1%, respectively; Fig. S1). Individuals who identified as wildlife advocates, animal rights advocates, and conservationists indicated greater support for reintroduction in their voting intentions (89.4%, 90.4%, and 87.6%, respectively) compared to those who did not (70.5%, 70.8%, and 74.7%, respectively; Fig. 4).

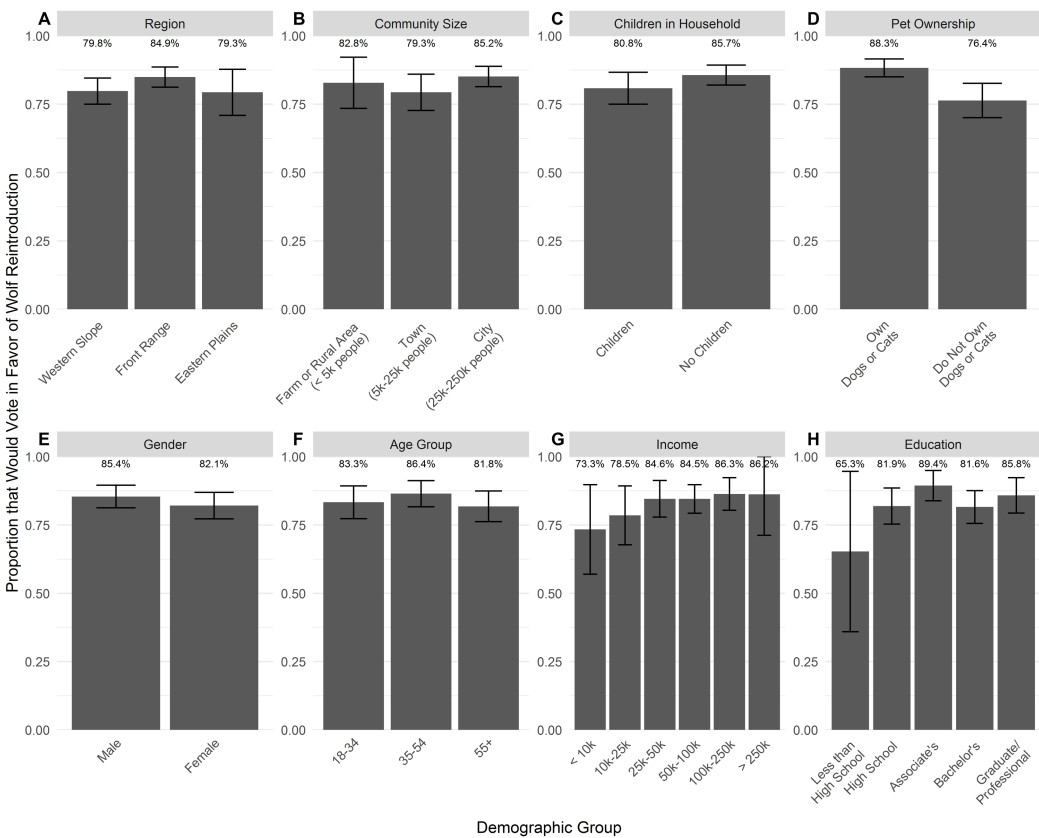

**Figure 3** Voting intentions related to wolf reintroduction by (A) region of Colorado, (B) community size, (C) presence of children in the household, (D) pet ownership, (E) gender, (F) age group, (G) income and (H) education. Bars depict the proportion of each group in favor of reintroduction (unweighted for region, weighted by region within other groups), with 95% confidence intervals. Non-binary genders not represented due to insufficient data (*n* = 5).

## Support for wolf management options

Generally, respondents were split on each of the proposed management options, but some relevant differences emerged (Fig. 5). Slightly more than 50% of Coloradans believed the following management options were slightly, moderately, or highly acceptable: limit the number of wolves if they cause declines in deer and elk populations in certain areas; compensate landowners for loss of livestock caused by a wolf; and, use a portion of state hunting and fishing license dollars to compensate landowners for loss of livestock caused by a wolf. Slightly less than 50% of Coloradans believed that the following options were acceptable: allow a recreational hunt of wolves once they have reached a certain population size that exceeds recovery goals; and, use a portion of state tax dollars to compensate landowners for loss of livestock caused by a wolf. Approximately 50% believed that capturing and lethally removing a wolf if it is known to have caused loss of livestock was acceptable.
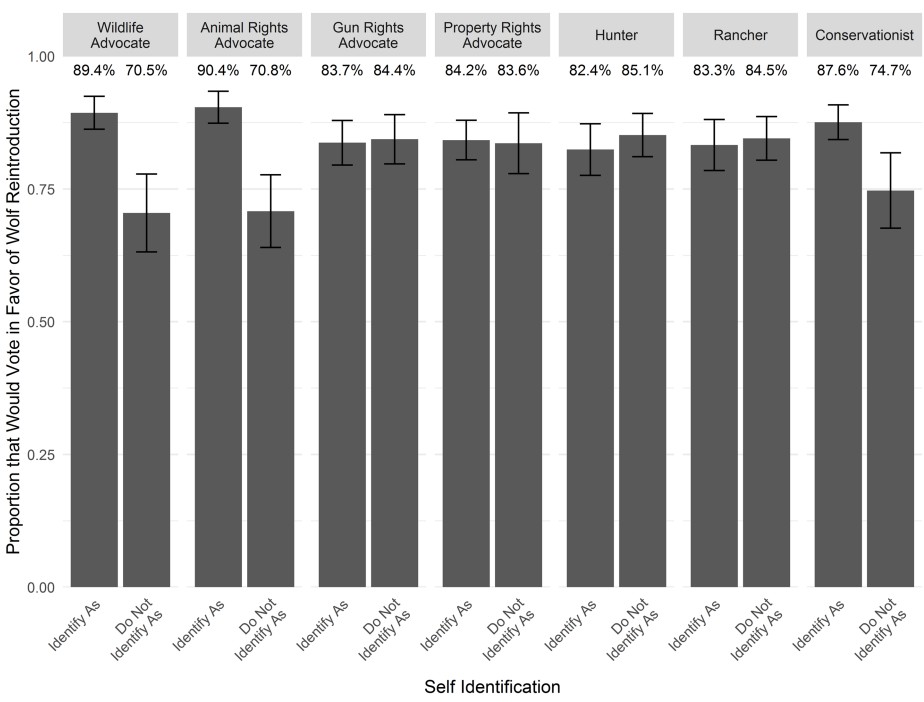

**Figure 4 Voting intentions related to self-identification with advocacy and interest groups.** Bars depict the proportion of individuals who identified with a group (a slight amount, a moderate amount or a great deal) who would vote in favor of reintroduction (weighted by region), with 95% confidence intervals.

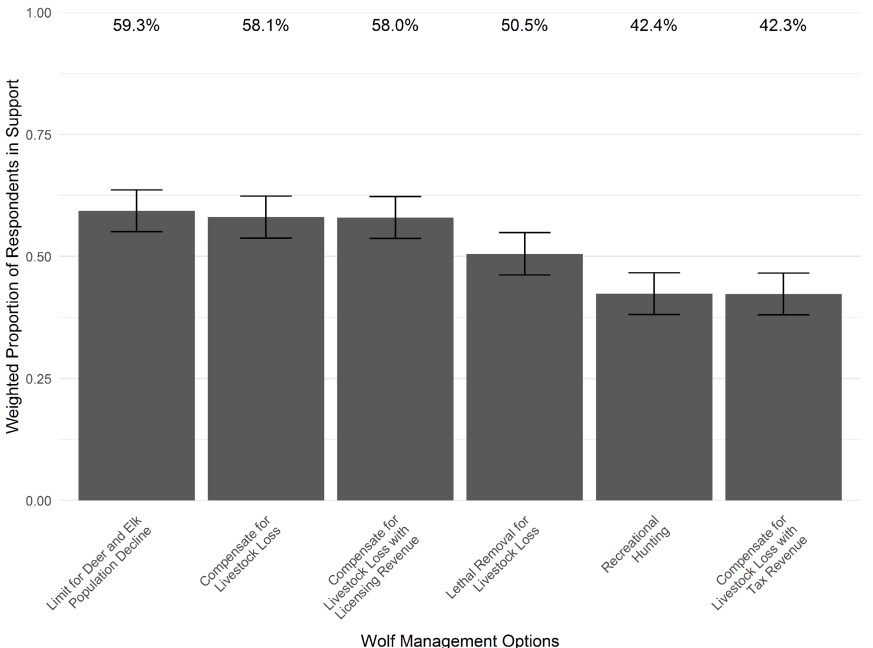

**Figure 5 Support for various wolf management options, should wolves be reintroduced to Colorado.** Bars depict the proportion of respondents who rate each management option at slightly, moderately, or highly acceptable (weighted by region), with 95% confidence intervals.

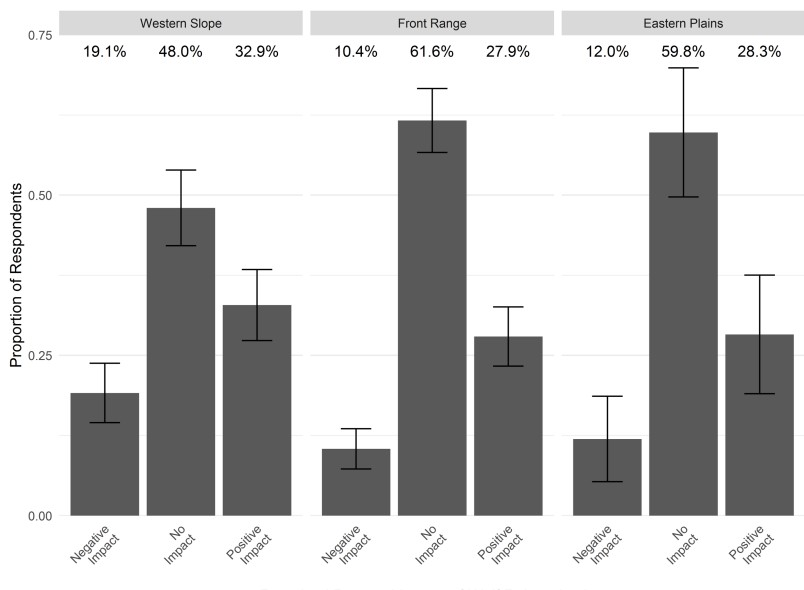

**Figure 6 Perceived impact on personal livelihoods or quality of life should wolves be reintroduced, by region of Colorado.** Bars depict the proportion of respondents within each region who report expectations of no impact, or expectations of slight, moderate, or strong negative or positive impacts (unweighted), with 95% confidence intervals. 

## Perceived impacts on livelihoods and quality of life

Responses to the question of how wolf reintroduction would impact personal livelihoods or quality of life showed that 28.6% of Coloradans would expect wolves to impact them positively, 11.7% would expect wolves to impact them negatively, and 59.7% would expect no impact. More residents on the Western Slope (19.1%) believed wolves would impact them negatively compared to the Front Range (10.4%; Fig. 6). Correspondingly, fewer residents on the Western Slope (48.0%) believed that wolves would have no impact compared to the Front Range (61.6%).

Our coding of open-ended responses revealed a variety of different reasons for why respondents believed wolf reintroduction would positively or negatively impact their livelihoods or quality of life (Fig. 7; Table S4 provides detailed descriptions and examples quotes for each coded theme). The percentages of respondents who mentioned each theme reported are out of the total number of respondents ($n = 320$) who indicated that wolves would impact them.

The most common way respondents believed wolves would positively impact their lives was through restoring balance to ecosystems and thereby enhancing ecosystem health. This positive impact was mentioned by 19.4% of the respondents who reported that wolves would impact them. For example, one respondent wrote, "Bringing wolves back into the ecosystem of Colorado would result in a healthier balance to nature." The second most commonly reported positive impact was seeing or hearing wolves in the wild (15.6%). Respondents discussed how they would enjoy encountering a wolf, and many even reported that they would spend more time outdoors to try to see wolves. Respondents also

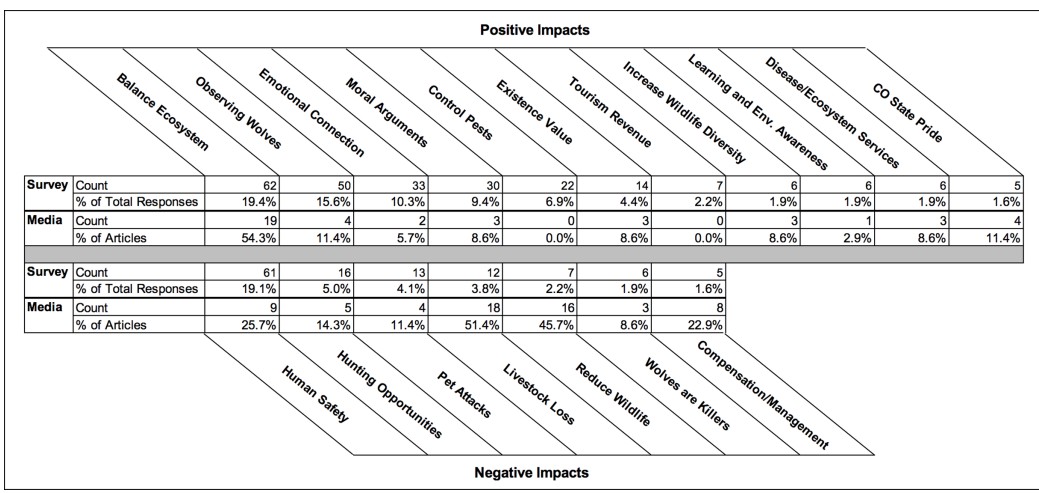

**Figure 7 Percentage of media articles and survey respondents reporting positive and negative themes related to the impacts of wolf reintroduction.**

discussed their emotional or cultural connection to wolves or their belief that wolves were "beautiful" or "majestic" (10.3%)." Some respondents (9.4%) provided moral arguments for reintroduction, including: wolf reintroduction can help correct past wrongs; wolves should have the right to live where they used to; humans should learn to coexist with nature; and, preserving a species is the right thing to do. Another reason reported among 6.9% of respondents was that wolves would reduce pest populations, particularly rodents and deer. Other, less commonly reported reasons included: respondents valued the existence of wolves now or for future generations (4.4%); wolf reintroduction would increase tourism (2.2%); wolves would increase the diversity or abundance of wildlife in Colorado (1.9%); wolves would help people learn about or appreciate nature (1.9%); wolves would help regulate disease, reduce the number of sick ungulates, or provide other ecosystem services to people (1.9%); and wolf reintroduction would enhance Colorado pride or make Colorado a better place to live (1.6%; Fig. 7; Table S4).

Those who indicated that wolves would negatively impact their lives most commonly reported that they were worried about wolves posing a threat to human safety or wandering into residential areas and causing conflict with humans (19.1%; Fig. 7; Table S4). Respondents also mentioned being concerned about wolves reducing hunting opportunities (5.0%), harming pets (4.1%), and threatening the livelihoods of ranchers through livestock depredation (3.8%). Some respondents (2.2%) believed wolf reintroduction would lead to a reduction in wildlife (e.g., elk and deer) abundance. Finally, some respondents (1.9%) mentioned concerns that wolves were "killers" or "cruel" animals, while others (1.6%) were concerned that wolves would be poorly managed or difficult to manage. For example, several respondents discussed how wolves were managed in other states and were concerned about ranchers not being fairly compensated for livestock losses from wolf predation.

## Media analysis

News articles discussing wolf reintroduction in Colorado included many of the same themes identified within open-ended survey responses (Fig. 7; Table S4). In the 35 articles analyzed, negative impact themes from survey responses were emphasized more frequently compared to positive impact themes; for example, 20 articles mentioned two or more negative impact themes, while 10 articles mentioned two or more positive impact themes. The average number of positive themes per news article was 1.2 (range 0–6), while the average number of negative themes was 1.8 (range 0–5).

The most commonly reported negative impact theme discussed in the media was concern about livestock depredation, mentioned in 51.4% of media articles. While a reduction in hunting opportunities was only explicitly discussed in 14.3% of articles, many (45.7%) brought up potential reductions in wildlife abundance (particularly deer and elk). News articles less often discussed the perceived threats of wolves to human safety (25.7%) and pets (11.4%), the difficulty of compensation and management (22.9%), and perceptions of wolves as killers (8.6%).

The most commonly reported positive impact theme discussed in the media (reported in 54.3% of the articles) was the possibility of wolves restoring balance to ecosystems and enhancing ecosystem health. Some articles mentioned the desire to see or hear wolves in the wild (11.4%); wolves being an important part of state pride (11.4%); the potential positive effects of wolves controlling disease in wildlife populations or providing other ecosystem services to humans (8.6%); moral arguments for the return of wolves (8.6%; e.g., "it's the right thing to do"); the existence value of wolves (8.6%); and wolves leading to increases in wildlife abundance or diversity (8.6%). Fewer articles discussed perceived emotional or cultural connections to wolves (5.7%) and opportunities for wolves to enhance environmental learning and awareness (2.9%). Control of pests and boosts in tourism with wolf watching were not discussed in the reviewed media articles.

While consistent themes were present in both survey responses and media, they were disproportionate in representation. Concerns about the threat wolves pose to livestock were expressed in only 3.8% of survey responses, but were mentioned in 51.4% of media articles (Fig. 7). The potential for wolves to provide balance to ecosystems was mentioned in 54.3% of media articles compared to only 19.4% of survey responses. Other arguments about the positive impacts of wolf reintroduction identified from surveys, such as emotional connections to wolves, reductions in pest populations, and increases in tourism, were rarely or never mentioned in the media.

Outside of the themes emerging from survey responses, media articles also often discussed the following negative impacts or concerns associated with reintroduction: the fact that the Colorado Parks and Wildlife Commission had voted against reintroduction in the past; the potential for wolves to disrupt ecosystems; the belief that making decisions about wildlife management through ballot initiatives is problematic; and the ability of wolves to recolonize Colorado naturally, potentially negating the need for reintroduction altogether. This last point was particularly salient in media articles

from January 2020, given that a wolf pack was documented as naturally moving into the state at that time. Additional arguments in support of wolf reintroduction that the media brought up were that the majority of Coloradans were in favor of wolf reintroduction, reintroduction would connect the entire North American wolf population from Mexico to the Arctic, and that the ballot initiative would be a democratic approach for allowing the majority of the public to have a say in decision-making about wildlife.

## DISCUSSION

There is a high level of public support for wolf reintroduction in Colorado in advance of the proposed ballot initiative. Our results suggest that approximately 84% of the population would vote for wolf reintroduction, which is over 13 percentage points greater than statewide estimates from 1994 (*Pate et al., 1996*) and 18 percentage points greater than estimates from 2001 (*Meadow et al., 2005*). While differences in participant recruitment methods demand caution in making direct comparisons to the earlier surveys, such increased support would be in line with changing public values towards wildlife in the state, which have been documented in other studies (*Manfredo et al., 2017*, *2019*). *Manfredo et al. (2019)* discuss how modernization has led to an increase in anthropomorphism of wildlife, contributing to a cultural shift among the public from domination (i.e., wildlife should be managed for human use) to mutualist (i.e., wildlife is more human-like and should have rights like humans) values toward wildlife. Such mutualist values have been associated with greater support for wolf conservation (*Hermann, Voß & Menzel, 2013*; *Dietsch, Teel & Manfredo, 2016*). It is likely that this growing emphasis on mutualist values among Coloradans is also reflected in our findings regarding the relatively low level of public support for recreational hunting of wolves.

We found that, in general, there were similar levels of support across many of the demographic and social identity variables we measured. We found slightly lower levels of support among individuals who identified strongly as ranchers and hunters, although the majority of respondents who identified as such still indicated they would vote in support of reintroduction. Support for wolf reintroduction was slightly higher among pet owners and individuals who identified as wildlife advocates, animal rights advocates, and conservationists. These results support previous research by *Bruskotter, Vaske & Schmidt (2009)*, who found that social identification with relevant stakeholder or interest groups can strongly affect people's attitudes and beliefs towards a species, which can in turn influence support for wildlife management options.

Our analysis revealed a diversity of potential positive impacts that the public believed would occur from wolf reintroduction. The most commonly reported positive impact from wolf reintroduction was that wolves would restore balance to ecosystems or enhance ecosystem health. This was also the most common argument in favor of wolf reintroduction reported by the media. These findings support previous literature suggesting that the preservation of ecosystem health is often a goal for conservation initiatives (*Nelson, 2009*) and that wolves and other predators can exert a positive

"top-down" influence in ecological communities (*Estes et al., 2011*; *Vucetich, Nelson & Peterson, 2012*). Given the importance placed on restoring ecosystem health by the public, ecological studies will be needed to evaluate the degree to which wolves could achieve this outcome in Colorado. While observational studies have suggested cascading effects of wolves in the Greater Yellowstone Ecosystem (*Ripple & Beschta, 2012*), the reintroduction of wolves alone may not be able to fully reverse the impacts of the removal of wolves on ecosystems (*Marshall, Hobbs & Cooper, 2013*), and it remains unclear to what extent wolves might have significant positive ecological effects outside of the protection provided by National Parks (*Mech, 2012*).

Additional perceived positive impacts of wolf reintroduction raised by the public included the opportunity to view wolves, a reduction in pest populations, moral arguments (e.g., "it's the right thing to do"), and emotional connections to wolves. Interestingly, these arguments were rarely covered by the media. Rather, media coverage of potential wolf reintroduction focused more on themes related to the negative impacts of wolves rather than the positive impacts identified by the public in the survey data. Among the different negative impacts brought up by survey respondents, media coverage focused most on livestock depredation and reductions in wildlife (e.g., deer and elk) abundance. These results provide further evidence to previous research, which found that media coverage of wolves between 1999 and 2008 in the United States and Canada was primarily negative (*Houston, Bruskotter & Fan, 2010*). Our findings also support studies on media coverage of large carnivores more broadly; for example, research suggests that local media (such as the news sources we reviewed) tends to focus on the threats of carnivores to local livelihoods (*Jacobson et al., 2012*; *Killion et al., 2019*; *Sadath, Kleinschmit & Giessen, 2013*). This more negative coverage by local journalists may be due to their awareness of vocal opposition from certain stakeholders with concerns about human-carnivore conflict or a broader negativity bias in media coverage (*Soroka, Fournier & Nir, 2019*). Research suggests that in general, news coverage of current affairs tends to be predominantly negative, possibly because the average person is more physiologically activated by negative rather than positive stories (*Soroka, Fournier & Nir, 2019*).

Media focus on possible negative impacts could have several implications for the wolf reintroduction initiative in Colorado. Studies suggest that such coverage could potentially reduce support for wolf reintroduction. For example, attitudes towards wolf reintroduction in the Adirondacks became more negative following negative media coverage (*Enck & Brown, 2002*), and media amplified perceived black bear-related risks among the public in New York (*Gore & Knuth, 2009*). In addition, the focus of media coverage on a limited number of the breadth of possible impacts we identified—both positive and negative—has the potential to limit the complexity of the discourse surrounding wolves. This could result in people believing that groups with opposing perspectives on wolf reintroduction have homogenous points of view. Social psychology research suggests that such perceptions of "outgroup homogeneity" (i.e., the belief that everyone in the opposing group is the same) can increase hatred towards outgroups, enhance social conflict, and pose a barrier to collaboration and reconciliation
(*Čehajić-Clancy et al., 2016*). Given the intensity of stakeholder conflict associated with wolf reintroduction and management (*Nie, 2002*), developing effective collaborative solutions will require demonstrating the complexity of all the diverse perspectives on the issue, rather than focusing on a single argument for or against reintroduction (*Madden & McQuinn, 2014*).

Our analysis revealed several concerns held by the public about wolf reintroduction that will have to be addressed through stakeholder engagement processes and outreach, education, and incentive programs if wolf reintroduction is mandated. The most common concern expressed by the public in our survey was that wolves would pose a threat to human safety. Given that wolf attacks on humans are rare, with only two human fatalities in the whole of North America attributed to wolves in the past century (*Linnell & Alleau, 2016*), our findings suggest that outreach is needed to inform the public about the actual risks of wolf attacks and how to reduce the risk of wolf attacks on pets. Furthermore, similar to the findings of prior studies (*Enck & Brown, 2002*; *Pate et al., 1996*), respondents discussed concerns about wolf depredation on livestock and a reduction of hunting opportunities. If wolf reintroduction occurs in Colorado, these concerns could be addressed through a wolf management plan that is developed using participatory approaches that involve all affected stakeholders, reduce social conflict, and facilitate consensus building (*Madden & McQuinn, 2014*). Such a process should especially seek to include and address the concerns of residents on the Western Slope, who live where wolves would likely be reintroduced, and who were more likely to indicate that wolves would have negative impacts on their livelihoods compared to residents in the Front Range.

Our study builds on the growing body of literature that utilizes online sampling as a promising alternative to traditional approaches for conducting public surveys. Online recruitment may, however, create some bias towards individuals with high technology awareness (*Keeter & McGeeney, 2015*). Thus, future surveys using additional methodologies, such as mail, in-person, or phone-based recruitment, would be useful to supplement our results. Alternative sampling designs may be particularly important for understanding the perspectives of populations that are less engaged with technology, such as older adults or rural residents. In addition, a diverse complement of qualitative and quantitative work will be needed to gain the necessary depth to understand the perspectives of key stakeholder groups, such as hunters and ranchers, who may be more directly impacted by wolf reintroduction.

## CONCLUSIONS

Our findings suggest a high degree of social tolerance for wolf reintroduction in Colorado across geographies and demographics. We find that the Colorado public has a wide range of reasons for supporting wolf reintroduction, including promoting ecosystem balance, wolf viewing opportunities, emotional connections to the species, and moral arguments. However, we also find that a portion of the public believes that wolves would negatively impact their livelihoods because of concerns about the safety of people and pets, loss of hunting opportunities, and potential wolf predation on livestock. These concerns—particularly those related to livestock losses—are being strongly reflected

in media discourse. Although the majority of Colorado residents support wolf reintroduction, we emphasize that it is critical to listen to the voices of those who are directly negatively impacted by wolf reintroduction. While livestock producers in the state are relatively small in number, the potential for direct negative impacts on their livelihoods is important to consider and adequately address. The negative impact on the livelihoods of ranchers and individuals reliant on the big game hunting industry may be disproportionately high compared to the positive impacts on the rest of the state. If wolf reintroduction occurs, an intensive stakeholder engagement process that involves a diversity of potentially affected groups in collaborative decision-making is needed to address these concerns and reduce social and human-wildlife conflict. Furthermore, public outreach is needed to share scientific findings regarding the potential impacts of wolves and convey a nuanced, accurate portrayal of the diversity of possible positive and negative impacts of wolf reintroduction in Colorado.

## ACKNOWLEDGEMENTS

We would like to thank the participants in our survey as well as an anonymous reviewer and Mr. Benjamin Ghasemi for their thoughtful reviews of this manuscript.

### Funding

The work was supported by a grant from the Colorado State University Pre-Catalyst for Innovative Partnerships Program. The funders had no role in study design, data collection and analysis, decision to publish, or preparation of the manuscript.

### Grant Disclosures

The following grant information was disclosed by the authors:
Colorado State University Pre-Catalyst for Innovative Partnerships Program.

### Competing Interests

The authors declare that they have no competing interests.

### Author Contributions

- Rebecca Niemiec conceived and designed the experiments, performed the experiments, analyzed the data, prepared figures and/or tables, authored or reviewed drafts of the paper, and approved the final draft.
- Richard E.W. Berl analyzed the data, prepared figures and/or tables, authored or reviewed drafts of the paper, and approved the final draft.
- Mireille Gonzalez performed the experiments, analyzed the data, prepared figures and/or tables, and approved the final draft.
- Tara Teel conceived and designed the experiments, performed the experiments, analyzed the data, authored or reviewed drafts of the paper, and approved the final draft.
- Cassiopeia Camara analyzed the data, prepared figures and/or tables, and approved the final draft.

- Matthew Collins analyzed the data, prepared figures and/or tables, and approved the final draft.
- Jonathan Salerno conceived and designed the experiments, authored or reviewed drafts of the paper, and approved the final draft.
- Kevin Crooks conceived and designed the experiments, authored or reviewed drafts of the paper, and approved the final draft.
- Courtney Schultz conceived and designed the experiments, authored or reviewed drafts of the paper, and approved the final draft.
- Stewart Breck conceived and designed the experiments, authored or reviewed drafts of the paper, and approved the final draft.
- Dana Hoag conceived and designed the experiments, authored or reviewed drafts of the paper, and approved the final draft.

### Human Ethics

The following information was supplied relating to ethical approvals (i.e., approving body and any reference numbers):

Final survey and administration procedures were approved by Colorado State University's Institutional Review Board (protocol #19-8942H).

### Data Availability

The raw survey data and code are available in the Supplemental Files.

### Supplemental Information

Supplemental information for this article can be found online at http://dx.doi.org/10.7717/peerj.9074#supplemental-information.

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
