# Peer review of "Public perspectives and media reporting of wolf reintroduction in Colorado"

_PeerJ, doi:10.7717/peerj.9074_

## Round 0.1 · original submission · Minor Revisions

Both referees have some concerns about the scope of claims made in the article. One referee notes that the number of media articles is small and thinks that it is problematic to compare them to the survey results in the way that you do. The other referee is concerned about claims about theory about which the empirical results don't actually say anything. Please provide a letter with detailed responses to the referee comments together with your resubmission.

Reviewer 1 ·

Basic reporting

This is a very well written and organized paper. I found it very accessible and very few errors or omissions.

Experimental design

I found the research questions to be well defined and logical. My only concern relates to the size of the sample if media articles (question #4). With only 15 articles collected I found this too be too small a sample to conduct a framing analysis. I also found it a bit odd that the media articles were coded based on the findings of the survey.

Validity of the findings

Results were well integrate into the existing literature and the discussion was well written. As I mentioned above,collecting only 15 articles suggests that the reintroduction of wolves was not a salient media subject and therefore, while informative, caution should be taken when comparing to the survey results. In other words, i don't think there is enough here to conduct a comparative analysis with the survey results.

Additional comments

I feel that the number of media articles is not sufficient enough to conduct any real analysis. At the same time, I do think it is informative to situate the media coverage when discussing practical implications. I would suggest dropping question #4 but including some of the findings from the articles in your discussion/implications.

·

Basic reporting

The article is written in a clear and concise language and is methodologically sound. The authors have done a credible job in surveying public opinions and analyzing media content regarding the wolf reintroduction issue.

There are a couple of important omissions with regard to references. Moreover, the way the figures are displayed can be improved. Below are my suggestions:

Abstract:

• Add some information about your sampling and sample size, e.g., the number of respondents and representativeness of the sample.
• Similarly, add some information about the number of newspapers analyzed and the timeframe

Introduction:

• Line 86: One important omission is the results of the survey by Meadow et al. (2005). Please include this article in your review:
https://doi.org/10.2193/0091-7648(2005)33[154:TIOPAO]2.0.CO;2


Results:

• L. 289: The second “responses” should be removed.
• L. 365: “increase in tourism” instead of “increased in tourism.”

Experimental design

Materials & Methods:

• L 141-144: Three regions are mentioned with 200 samples planned for each; however, the total sample size is 734 (L. 168). Please clarify why the sample size exceeds the planned number (3x200=600). Also, provide some information on what region classification was used (e.g., Colorado Parks and Wildlife management areas). It is important to provide this information since the study claims to survey representative, state-wide opinions.
• L. 169-196: The use of online panels is getting more common in human dimensions of wildlife research and similar areas. The authors have done a good job of elaborating on the pros and cons of such a method of data collection. However, these two paragraphs seem to be too lengthy, given that the audience is familiar with these issues. I suggest the authors condense this content to a few sentences mentioning the main points and citations.
• Instead of the above content about online panels, it is worthwhile to include the information regarding the “Weighting of Survey Data” in the main text, instead of the supplementary material.

Results:

• Some of the figures 2-6 can be combined into one figure. For instance, Figures 2 and 3 can be combined. In this way, the information becomes more accessible to the reader. Moreover, the authors could free up some space and add Figure S-1 to the main text. Figure S-1 contains information that deserves being in the main text.
• Some of the percentage values are reported with Confidence Intervals in the text and some are not. I suggest the authors be consistent in the format of reporting these values.

Validity of the findings

Discussion:

• L. 380: Compare your results also with the results from the Meadow et al.’s survey:
https://doi.org/10.2193/0091-7648(2005)33[154:TIOPAO]2.0.CO;2
• L. 388-389: Since value orientations were not included in this study, this conclusion should be made with caution. For instance, “It is likely that ...” or “We speculate that …”
• L. 411-423: Briefly speculate why media coverage is more about the negative outcomes.
• L. 454: Briefly reiterate why West Slope residents are likely to be more suspicious.
• L. 455: Briefly add the possible bias due to collecting data through an online panel.
• On the possible effects of media, it is worth mentioning the ‘social amplification of risk’ literature. For example, see Gore & Knuth (2009):
https://doi.org/10.2193/2008-343

Additional comments

The manuscript reports the results of a very timely survey on public support for wolf reintroduction in Colorado. This is currently a hot topic in Colorado and the United States with important implications for wildlife management and conservation. It is methodologically sound with sufficient information to be replicated. However, the authors could have done a better job by theorizing the possible drivers of public attitudes toward the wolf reintroduction. In the current format, the manuscript does not contribute to the theory, however, it has important and timely information to address policy and practice in wildlife management and conservation.

---

## Round 0.2 · accepted · Accept

You have done a thorough job of responding to the referee comments.